# Reinforcement Learning from Imperfect Demonstrations

## Abstract

Robust real-world learning should benefit from both demonstrations and inter-action with the environment. Current approaches to learning from demonstra-tion and reward perform supervised learning on expert demonstration data and use reinforcement learning to further improve performance based on reward from the environment. These tasks have divergent losses which are difficult to jointly optimize; further, such methods can be very sensitive to noisy demonstrations. We propose a unified reinforcement learning algorithm, Normalized Actor-Critic (NAC), that effectively normalizes the Q-function, reducing the Q-values of ac-tions unseen in the demonstration data. NAC learns an initial policy network from demonstration and refines the policy in a real environment. Crucially, both learn-ing from demonstration and interactive refinement use exactly the same objective, unlike prior approaches that combine distinct supervised and reinforcement losses. This makes NAC robust to suboptimal demonstration data, since the method is not forced to mimic all of the examples in the dataset. We show that our unified rein-forcement learning algorithm can learn robustly and outperform existing baselines when evaluated on several realistic driving games.

## 1 Introduction

Deep reinforcement learning (RL) has achieved significant success on many complex sequential decision-making problems. However, RL algorithms usually require a large amount of interaction with an environment to reach good performance (Kakade et al., 2003); initial performance may be nearly random, clearly suboptimal, and often rather dangerous in real-world settings such as autonomous driving. Learning from demonstration is a well-known alternative, but typically does not leverage reward, and presumes relatively small-scale noise-free demonstrations. We develop a new robust algorithm for Reinforcement-Learning-from-Demonstrations (RLfD) that can learn value and policy functions from imperfect demonstration data as well as environmental reward signals.

Recent efforts toward policy learning which does not suffer from a suboptimal initial performance generally perform RLfD by leveraging an initial phase of supervised learning and/or auxiliary task learning. Several authors have shown that demonstrations can speed RL by mimicking expert data with a temporal difference regularizer (Hester et al., 2017) or via gradient-free optimization (Ebrahimi et al., 2017), yet these methods presume near-optimal demonstrations. Shelhamer et al. (2016) and Jaderberg et al. (2016) obtained better initialization via auxiliary task losses (e.g., pre-dicting environment dynamics) in a self-supervised manner; performance is still initially random with these approaches.

Our approach, Normalized Actor-Critic (NAC), is based on the maximum entropy reinforcement learning framework (Toussaint, 2009; Haarnoja et al., 2017b; Schulman et al., 2017). Our method performs RLfD using exactly the same algorithm to process both offline demonstration data and online experience, without any additional supervised loss function. This enables robust learning even from very corrupted (or even partially adversarial) demonstrations that contains $(s, a, r, s')$, since no assumption on the optimality of the data is required. This is enabled by a normalized formulation of the soft Q-learning gradient, which can also be regarded as a variant of the policy gradient.

We evaluate our result in realistic 3D simulated environments, Torcs and Grand Theft Auto V (GTA V), using access only to raw image inputs and game-generated rewards. Experimental results con-

firm that our approach outperforms previous approaches on driving tasks with only a modest amount of demonstration and that significant noise in the demonstrations are well-tolerated, as our method utilizes rewards rather than simply imitating demonstrated behaviors.

## 2 PRELIMINARIES

In this section, we will briefly review the reinforcement learning techniques that our method builds on, including maximum entropy reinforcement learning and the soft Q-learning.

### 2.1 MAXIMUM ENTROPY REINFORCEMENT LEARNING

The reinforcement learning problem we consider is defined by a Markov decision process(MDP) (Thie, 1983). Specifically, the MDP is characterized by a tuple $< \mathbb{S}, \mathbb{A}, \text{R}, \text{T}, \gamma >$, where $\mathbb{S}$ is the set of states, $\mathbb{A}$ is the set of actions, $R(s, a)$ is the reward function, $T(s, a, s') = P(s'|s, a)$ is the transition function and $\gamma$ is the reward discount factor. The agent interacts with the environment by taking action at some state, receiving the reward, and transiting to the next state.

In the conventional reinforcement learning setting (Sutton & Barto, 1998), the goal of an agent is to learn a policy $\pi_{std}$, such that it maximizes the future discounted reward:

$$\pi_{std} = \text{argmax}_\pi \sum_t \gamma^t \mathbb{E}_{s_t, a_t \sim \pi(s, a)}[R(s_t, a_t)] \tag{1}$$

Maximum entropy policy learning (Ziebart, 2010; Haarnoja et al., 2017b) uses an entropy augmented reward, so that the optimal policy will not only optimize for discounted future rewards, but also maximize the discounted future entropy of the action distribution:

$$\pi_{max-ent} = \text{argmax}_\pi \sum_t \gamma^t \mathbb{E}_{s_t, a_t \sim \pi(s, a)}[R(s_t, a_t) + \alpha H(\pi(\cdot|s_t))] \tag{2}$$

where $\alpha$ is a weighting term to balance the importance of the entropy. Unlike previous attempts of only adding the entropy term at a single time step, maximum entropy policy learning maximizes the discounted future entropy over the whole trajectory. Maximum entropy reinforcement learning has many benefits, such as better exploration in multi-modal problems, and establishing connections between Q learning and the actor-critic method (Haarnoja et al., 2017b; Schulman et al., 2017).

### 2.2 SOFT VALUE FUNCTIONS

Since the maximum entropy RL paradigm augments the reward with an entropy term, the definition of the value functions naturally changes to

$$Q_\pi(s, a) = r(s_0, a_0) + \mathbb{E}_{(s_1, a_1, \ldots) \sim \pi(s)} \sum_{t=1}^{\infty} \gamma^t (R(s_t, a_t) + \alpha H(\pi(\cdot|s_t))) \tag{3}$$

$$V_\pi(s) = \mathbb{E}_{(s_1, a_1, \ldots) \sim \pi(s)} \sum_{t=0}^{\infty} \gamma^t (R(s_t, a_t) + \alpha H(\pi(\cdot|s_t))) \tag{4}$$

where $\pi$ is some policy that value functions evaluate on. Given the state-action value function $Q^*(s, a)$ of the optimal policy, Ziebart (2010) show that the optimal state value function and the optimal policy could be expressed as:

$$V^*(s) = \alpha \log \sum_a \exp(Q^*(s, a)/\alpha) \tag{5}$$

$$\pi^*(a|s) = \exp((Q^*(s, a) - V^*(s))/\alpha) \tag{6}$$

### 2.3 SOFT Q-LEARNING AND POLICY GRADIENT

With the entropy augmented reward, one can derive the soft versions of Q learning and policy gradient (Haarnoja et al., 2017a). The soft Q learning gradient is given by

$$\nabla_\theta Q_\theta(s, a)(Q_\theta(s, a) - \hat{Q}(s, a)) \tag{7}$$

where $\hat{Q}(s, a)$ is a bootstrapped Q-value estimate obtained by $R(s, a) + \gamma V_Q(s')$. Here, $R(s, a)$ is the reward received from the environment, $V_Q$ is computed from $Q_\theta(s, a)$ with Equation (5). We can also derive a policy gradient variant of this method, by parameterizing as the policy as $\log \pi_\theta(a|s) = Q_\theta(s, a) - V_\theta(s)$, which induces a policy gradient of the form

$$\mathbb{E}\left[\sum_{t=0}^{\infty} \nabla_\theta \log \pi_\theta(a_t|s_t)(\hat{Q}_\pi(s_t, a_t) - b(s_t)) + \alpha \nabla_\theta H(\pi_\theta(\cdot|s_t))\right] \tag{8}$$

where $b(s_t)$ is some arbitrary baseline (Schulman et al., 2017).

## 3 ROBUST LEARNING FROM DEMONSTRATION AND REWARD

Given a set of demonstrations that contains $(s, a, r, s')$ and the corresponding environment, an agent should perform appropriate actions when it starts the interaction, and continue to improve. Although a number of off-policy RL algorithms could in principle be used to learn directly from off-policy demonstration data, standard methods such as Q-learning can suffer from extremely poor performance when trained entirely on demonstration data, as shown in our experiments. The intuition behind this problem is that, if the Q-function is trained only on good data, it has no way to understand why the action in the data is good: it will assign it a high Q-value, but will not necessarily assign a low Q-value to other alternative actions. The framework of soft optimality provides us with a very natural mechanism to mitigate this problem by *normalizing* the Q-function over the actions.

Our approach, Normalized Actor-Critic (NAC), combines the soft Q-learning and soft policy gradient formulations described in the previous section to obtain a Q-function gradient that naturally reduces the Q-values of actions that were not observed along the demonstration. Put another way, without data to indicate otherwise, NAC will opt to follow the demonstration. This method is derived from the policy gradient in Equation (8), which introduces a normalization term $\nabla_\theta V(s)$ that reduces the value of unobserved actions. Since this method is still a well-defined RL algorithm without any auxiliary supervised loss, it is able to learn without bias in the face of low-quality demonstration data. We will first describe our algorithm, and then further discuss why it performs well when trained on off-policy demonstration data.

### 3.1 NORMALIZED ACTOR-CRITIC

We propose a unified learning from demonstration approach, which applies the normalized actor-critic updates to both off policy demonstrations and in-environment transitions. To better facilitate the comparison with the Q learning method, we re-parametrize the normalized actor-critic with Q function. As shown in the appendix A.1, we have the updates for the actor and the critic being:

$$\nabla_\theta J_{PG} = \mathbb{E}_{s, a \sim \pi_Q} \left[ (\nabla_\theta Q(s, a) - \nabla_\theta V_Q(s))(Q(s, a) - \hat{Q}(s, a)) \right] \tag{9}$$

$$\nabla_\theta J_V = \mathbb{E}_s \left[ \nabla_\theta \frac{1}{2}(V_Q(s) - \hat{V}(s))^2 \right] \tag{10}$$

where $V_Q$ and $\pi_Q$ are deterministic functions of $Q$: $V_Q(s) = \alpha \log \sum_a \exp(Q(s, a)/\alpha)$; $\pi_Q(a|s) = \exp((Q(s, a) - V_Q(s))/\alpha)$. $\hat{Q}(s, a), \hat{V}(s)$ are obtained by:

$$\hat{Q}(s, a) = R(s, a) + \gamma V_Q(s') \tag{11}$$

$$\hat{V}(s) = \mathbb{E}_{a \sim \pi_Q}[R(s, a) + \gamma V_Q(s')] + \alpha H(\pi_Q(\cdot|s)) \tag{12}$$

As discussed by Schulman et al. (2017); Haarnoja et al. (2017b), the method can be interpreted as either a Q-learning algorithm or a policy gradient method. The two interpretations lead to slightly different variants of the update equations. The soft Q-learning variant omits the $\nabla_\theta V(s)$ in Equation ((9)). In our normalized actor critic method, we emphasize the normalization effect from the $\nabla_\theta V(s)$ term; this term enables our method to learn from off-policy data. Our method also requires importance sampling to use off-policy data. As discussed in Section 3.2, our method critically relies on the $\nabla_\theta V(s)$ term to attain good performance on demonstration data, which means that, in order to formalize it as a policy gradient method, we should include importance weights. To be specific,

when estimating $\mathbb{E}_{(s,a)\sim\pi_Q}[f(s,a)]$, we estimate $\mathbb{E}_{(s,a)\sim\mu}[f(s,a)\beta]$, where $\beta = \min\left\{\frac{\pi_Q(a|s)}{\mu(a|s)}, c\right\}$ and $c$ is some constant that prevents the importance ratio from being too large.

Although the importance weights are needed to formalize our method as a proper policy gradient algorithm, we find in our empirical evaluation that the inclusion of these weights consistently reduces the performance of our method even on demonstrations. For this reason, our final algorithm does not use importance sampling.

We summarize the proposed method in Algorithm 1. Our method uses samples from the demonstrations and the replay buffer, rather than restricting the samples to be on policy as in standard actor-critic methods. Similar to DQN, we utilize a target network to compute $\hat{Q}(s,a)$ and $\hat{V}(s)$, which stabilizes the training process.

---

**Algorithm 1** The Normalized Actor-Critic Algorithm

$\theta$: **parameters for the rapid Q network,** $\theta'$: **parameters for the target Q network,** $\mathcal{D}$: **demonstrations collected by human or a trained policy network,** $T$: **target network update frequency,** $\mathcal{M}$: **replay buffer,** $k$: **number of steps to train on the demonstrations**
**for** step $t \in \{1, 2, ...\}$ **do**
    **if** t $\leq k$ **then**
        Sample a mini-batch of transitions from $\mathcal{D}$
    **else**
        Start from s, sample $a$ from $\pi$, execute $a$, observe $(s',r)$ and store $(s,a,r,s')$ in $\mathcal{M}$
        Sample a mini-batch of transitions from $\mathcal{M}$
    **end if**
    Update $\theta$ with gradient: $\nabla_\theta J_{PG} + \nabla_\theta J_V$
    **if** t mod T = 0 **then**
        $\theta' \leftarrow \theta$
    **end if**
**end for**

---

## 3.2 ANALYSIS OF THE METHOD

We provide an intuitive analysis in this section to reason why our method can learn from demonstrations while other reinforcement learning methods, such as Q-learning, can not. The states and actions in the demonstrations generally have higher Q-values than other states. Q-learning will therefore push up $Q(s,a)$ values in a sampled state $s$. However, if the values for the bad actions are not observed, then the Q-function has no way of knowing whether the *action* itself is good, or whether all actions in that *state* are good, so the demonstration action will not necessarily have a higher Q-value than other actions in the demonstration state. Comparing the actor update (Eq. (9)) of our method with the soft Q-learning (Eq. 7) update, our method includes an extra term in the gradient: $-\nabla_\theta V_Q(s)$. This term falls out naturally when we consider the method as a policy gradient algorithm, rather than a Q-learning algorithm (Haarnoja et al., 2017b). Intuitively, this term will decrease $V_Q(s)$ when increasing $Q(s,a)$ and vice versa, since $\nabla_\theta Q(s,a)$ and $-\nabla_\theta V_Q(s)$ have different signs. Since $V_Q(s) = \alpha \log \sum_a \exp(Q(s,a)/\alpha)$, decreasing $V_Q(s)$ will prevent $Q(s,a)$ from increasing for the actions that are not in the demonstrations.

## 4 RELATED WORK

Recently, deep reinforcement learning has achieved great success on many tasks. Notable examples include deep Q-learning (Mnih et al., 2015), deep visuomotor policy learning (Levine et al., 2016) and trust region policy optimization (Schulman et al., 2015). Other recent successes include expert action prediction in Go (Silver et al., 2016) and intelligent agents in Starcraft II and vizDoom (Vinyals et al., 2017; Kempka et al., 2016). In this section, we review the related work in deep RL as well as imitation learning.

### 4.1 LEARNING FROM DEMONSTRATION

Imitation learning can learn meaningful behaviors from expert demonstrations. For example, Xu et al. (2016); Bojarski et al. (2016) learn driving policies from human demonstrations. One popular paradigm of imitation learning is DAGGER (Ross et al., 2011), which improves an agent's performance with the expert's feedback in the loop. Recently, Ho & Ermon (2016); Wang et al. (2017); Ziebart et al. (2008) utilize an adversarial paradigm to further boost the performance of the behavior cloning method. Another popular paradigm is Inverse Reinforcement Learning (IRL) (Ng et al., 2000; Abbeel & Ng, 2004; Ziebart et al., 2008). IRL learns a reward model which explains the demonstrations as optimal behaviors. The reward function can be represented as the distance to a demonstrated trajectory, as a weighted combination of features.

Recently, Subramanian et al. (2016) showed the exploration strategy in an environment can be guided by human demonstrations. Taylor et al. (2011) directly transfer knowledge from expert demonstrations to the learning agent. Guided policy search can be seen as a form of imitation learning, where the expert is the optimal control policy (Levine & Koltun, 2013). To have a smoother transition from imitation learning to reinforcement learning, Hester et al. (2017) propose to use an objective that combines a supervised loss and a temporal difference loss. Both Hester et al. (2017) and our algorithm follow the Reinforcement Learning with Expert Demonstrations (RLED) framework (Chemali & Lazaric, 2015; Kim et al., 2013; Piot et al., 2014), where both rewards and actions are available in the demonstration. We follow the terminology from Hester et al. (2017) and use Learning from Demonstration to describe our task but rewards are available from the demonstrations in our setting. Different from previous works, our method does not require an explicit mechanism to determine expert data because it learns to maximize the reward using arbitrary data.

### 4.2 OFF-POLICY LEARNING

Reinforcement learning methods learn a policy while interacting with the environment. Some methods require the data being used to update the current policy $\pi_t$ to be roll-outs of $\pi_t$ in the environment, while other methods allow the updates to be computed from data generated by arbitrary policy. Those methods are classified as on-policy methods and off-policy methods respectively. The most well known off-policy method is Q-learning (Watkins & Dayan, 1992). Policy gradient (Sutton et al., 2000) methods require on-policy data, but recently some works have proposed variants of policy gradient that can learn from off-policy data (Gu et al., 2017; 2016). Off-PAC (Degris et al., 2012) is the first off-policy actor-critic method. Munos et al. (2016) proposed Retrace($\lambda$), a method that allows multiple step return to being used with off-policy updates. ACER (Wang et al., 2016) combines the advances of Off-PAC, Retrace($\lambda$), trust region update and leads to a stable and sample efficient actor-critic method. However, some methods, such as Retrace, require the off-policy data to be close to the current policy. The other category of off-policy methods consists of Q-learning methods, which we care to in our evaluation and find that they generally do not perform well on purely off-policy data.

### 4.3 MAXIMUM ENTROPY REINFORCEMENT LEARNING

Maximum entropy reinforcement learning has been explored in a number of prior works Todorov (2008); Toussaint (2009); Ziebart et al. (2008), including several recent works that extend it into a deep reinforcement learning setting Nachum et al. (2017); Haarnoja et al. (2017b). Our work builds on the connection between policy gradient and soft Q-learning discussed by Haarnoja et al. (Haarnoja et al., 2017b) and Schulman et al. (Schulman et al., 2017). However, unlike these prior works, we are specifically concerned with learning from demonstrations, and we show how a novel algorithm that combines elements of policy gradient and Q-learning is especially well suited for learning from demonstrations, while still optimizing the entropy-augmented maximum reward objective, without any explicit imitation loss.

## 5 RESULTS

We address several questions with our experiments: (1) Can NAC benefit from both demonstrations and rewards? (2) Is NAC robust to ill-behaved demonstrations? (3) Can NAC learn meaningful behaviors with a limited amount of demonstration? We compare our algorithm with DQfD

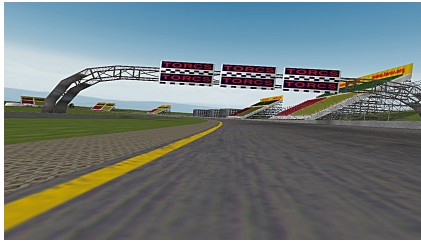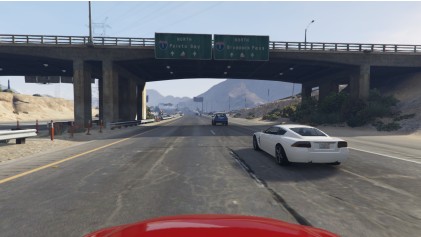

Figure 1: Sample frames from Torcs (left) and GTA (right).

(Hester et al., 2017), which has been shown to learn efficiently from demonstrations and to preserve performance while acting in an environment. Other baselines include supervised behavioral cloning method, Q-learning, soft Q-learning, and the version of our method with importance sampling weighting.

## 5.1 ENVIRONMENTS

We evaluate our algorithm on Torcs and Grand Theft Auto V (GTA) shown in Fig 1.

**Torcs**: Torcs is an open-source racing game that has been used widely as an experimental environment for driving. The goal of the agent is to drive as fast as possible on the track, while avoiding crashes. We use an oval two-lane racing venue in our experiments. The input to the agent is an $84 \times 84$ gray scale image. The agent controls the vehicle at 5Hz, and at each step, it chooses from a set of 9 actions which is a Cartesian product between {left, no-op, right} and {up, no-op, down}. We design a dense driving reward function that encourages the car to follow the lane and to avoid collision with obstacles. [1]

**GTA**: Grand Theft Auto is an action-adventure video game with goals similar in part to the Torcs game, but with a more diverse and realistic surrounding environment, including other vehicles, buildings, and bridges. The agent observes $84 \times 84$ RGB images from the environment. It chooses from 7 possible actions from {left-up, up, right-up, left, no-op, right, down} at 6Hz. We use the same reward function as in Torcs.

## 5.2 BASELINES

We compare our approach with the following baselines:

- **DQfD:** the method proposed by Hester et al. (2017). For the learning-from-demonstration phase, DQfD combines a hinge loss with a temporal difference (TD) loss. For the finetuning-in-environment phase, DQfD combines a hinge loss on demonstration and a TD loss on both the demonstration and the policy-generated data.

- **Q learning:** the classic DQN method (Mnih et al., 2015). We first train DQN with the demonstrations in a replay buffer and then finetune in the environment with regular Q learning. Similar to DQfD, we use a constant exploration ratio of 0.01 in the finetuning phase to preserve the performance obtained from the demonstration.

- **Soft Q learning:** similar to the Q learning baseline, but with an entropy regularized reward. This is the method proposed by Haarnoja et al. (2017a); Schulman et al. (2017).

- **Behavior cloning with Q learning:** the naive way of combining cross entropy loss with Q learning. First we perform behavior cloning with cross entropy loss on the demonstrations. Then we treat the logit activations prior the softmax layer as an initialization of the Q function and finetune with regular Q learning in the environment.

- **Normalized actor-critic with importance sampling:** the NAC method with the importance sampling weighting term mentioned in Sec 3.1. The importance weighting term is used to correct the action distribution mismatch between the demonstration and the current policy.

---

[1]$reward = (1 - \mathbb{1}_{damage}) \left[ (\cos \theta - \sin \theta - lane\_ratio) \times speed \right] + \mathbb{1}_{damage} [-10]$, where $\mathbb{1}_{damage}$ is an indicator function of whether the vehicle is damaged at the current state. $lane\_ratio$ is the ratio between distance to lane center and lane width. $\theta$ is the angle between the vehicle heading direction and the road direction.

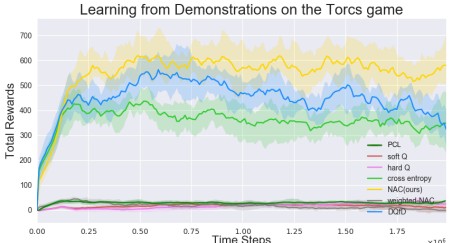 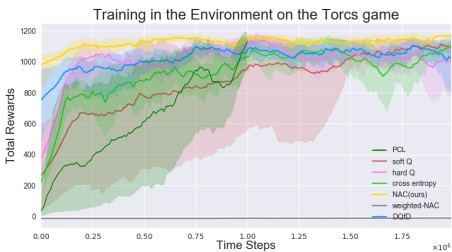

Figure 2: Performances on the Torcs game. The x-axis shows the training iterations. The y-axis shows the average total rewards. Solid lines are average values over 10 random seeds. Shaded regions correspond to one standard deviation. The left figure shows the performance for each agent when they only learn from demonstrations, while the right one shows the performance for each agent when they interact with the environments after learning from demonstrations. Our method consistently outperforms other methods in both cases.

- **Path Consistency Learning (PCL):** the PCL method that minimizes soft path consistency losses extracted from action sequences from both off-policy data and on-policy data. Different from original implementation Nachum et al. (2017), we only consider one step ahead the path.

### 5.3 COMPARISON TO BASELINES

We first compare our NAC method with all baselines on 300k transitions. The demonstration is collected by a trained Q-learning expert policy. We execute the policy in the environment to collect demonstrations. To avoid deterministic executions of the expert policy, we uniformly sample an action with probability 0.01.

To explicitly compare different methods, we show separate figures for performances on demonstrations and in environments. In Fig 2, we show that our method performs better than other methods on demonstrations. When we start finetuning, the performance of our method continues to increase and reaches peak performance faster than the baselines. In comparison, Q learning, PCL and soft Q learning can barely learn from demonstrations. DQfD (Hester et al., 2017) has similar behavior to ours but has lower performance. Behavior cloning learns well on demonstrations, but it has a significant performance drop while interacting with environments. All the methods can ultimately learn by interacting with the environment but only our method and DQfD start from a relatively high performance. Empirically, we found that the importance weighted NAC method does not perform as well as NAC. The reason might be the decrease in the gradient bias is not offset sufficiently by the increase in the gradient variance.

We also test our method on the challenging GTA environment, where both the visual input and the game logic are more complex. Due to the limit of environment execution speed, we only compare our method with DQfD. As shown in Fig. 3a, our method outperforms DQfD both on demonstration and in environment.

### 5.4 LEARNING FROM HUMAN DEMONSTRATIONS

For many practical problems, such as autonomous driving, we might have a large number of human demonstrations, but no demonstration available from a trained agent at all. In contrast to a scripted agent, humans usually act diversely, both in terms of multiple individuals (e.g. conservative players will slow down before a U-turn; aggressive players will not) and a single individual (e.g. a player may randomly turn or go straight at an intersection). We study how different methods perform with diverse demonstrations. To collect human demonstrations, we asked 3 non-expert human players to play TORCS for 3 hours each. Human players control the game with the combination of four arrow keys, at 5Hz, the same rate as the trained agent. The experiment is run for 800k training steps, as indicated from the experiments in Sec 5.3. In total, we collected around 150k transitions. Comparing with data collected from trained agent, the data is more diverse and the performance of the demonstration improves naturally when the players get familiar with the game. In Fig. 3b, we show the comparison among our NAC method, DQfD, and behavior cloning; we omit Q learning, soft Q learning, and the importance weighted NAC are shown to perform poorly in the demonstration

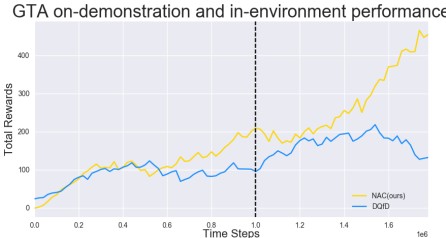
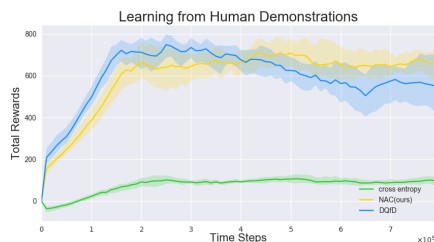

(a) The on-demonstration and in-environment performance of the NAC and DQfD methods on GTA. The vertical line separates the learning from demonstration phase and finetuning in environment phase. Our method consistently outperforms DQfD in both phases.

(b) Performances on the Torcs game with human demonstrations. DQfD performs well in the beginning, but overfits in the end. The behavior cloning method is much worse than NAC and DQfD. Our NAC method performs best at convergence.

Figure 3: Performance on GTA (left) and performance on Torcs with human demonstrations (right)

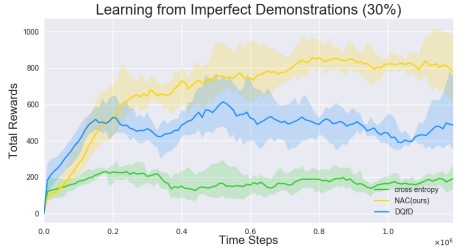
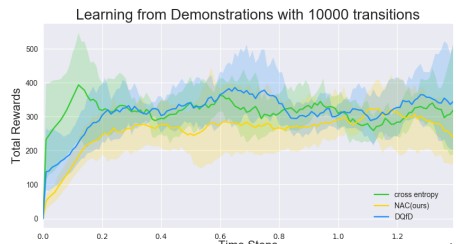

Figure 4: Left: Learning from imperfect data when the imperfectness is 30%. Our NAC method does not clone suboptimal behaviors and thus outperforms DQfD and behavior cloning. Right: Learning from a limit amount of demonstrations. Even with only 30 minutes (10k transitions) of experience, our method could still learn a policy that is comparable with supervise learning method. More results are available in appendix Fig. 5 and Fig. 6.

phase. We observe that the behavior cloning method performs much worse than the other two methods. DQfD initially is better than our method but later overfits quickly, which might be caused by the supervised hinge loss being harmful when demonstrations are not optimal. NAC does not assume either perfect or consistent demonstrations, and performs better than DQfD and behavior cloning at convergence.

## 5.5 EFFECTS OF IMPERFECT DEMONSTRATIONS

In the real world, collected demonstrations might be far from optimal. The human demonstrations above have already shown that imperfect demonstrations could have a large effect on performance. To study this phenomenon in a principled manner, we collect a few versions of demonstrations with varying degrees of noise. When collecting the demonstrations with the trained Q agent, we corrupt a certain percentage of the demonstrations by choosing actions regarding $(\min_a Q(s, a))$. The data corruption process is conducted while interacting with the environment; therefore, the error accumulated after the corrupted frames will also influence the quality of the dataset in a similar way as human demonstrations. We get 3 sets of $\{30\%, 50\%$ and $80\%\}$ percentage of imperfect data. In the left of Fig. 4, we show that our method performs quite well compared with DQfD and behavior cloning baselines. Supervised behavior cloning method is heavily influenced by the imperfect demonstrations. DQfD is also heavily affected, but not as severely as the behavior cloning. NAC is robust because it does not imitate the suboptimal behaviors.

## 5.6 EFFECTS OF DEMONSTRATION SIZE

In this section, we show comparisons among our method and baseline methods with different amounts of demonstration data. We use a trained agent to collect three sets of demonstrations which include 10k, 150k, and 300k transitions each. In the experiments we find that our algorithm per-

forms well when the amount of data is large and is comparable to supervised methods even with a limited amount of data. In Fig. 4(right), we show when there are extremely limited amounts of demonstration data (10k transitions or 30 minutes of experience), our method performs on par with supervised methods. In Fig. 6, we show the results for 150k and 300k transitions: our method out-performs the baselines by a large margin with 300k transitions. In summary, our method can learn from small amounts of demonstration data and dominates in terms of performance when there is a sufficient amount of data.

## 5.7 INTUITIVE ANALYSIS

In this section, we analyze the reason why other reinforcement learning methods can hardly learn from demonstrations while our method can. We use a concrete example to clarify and complement the mathematical discussion in Section 3.2. Suppose at state $s_i$, we observe the demonstrated behavior choosing action $a_1$, and receiving a positive reward of 1.0. The action space is $a_0, a_1, a_2$ for each state. Soft Q learning method takes an update of $\nabla Q(s_i, a_1)(Q(s_i, a_1) - \hat{Q}(s_i, a_1))$ for the parameter $\theta$ of the Q function, where $\hat{Q}(s_0, a_1) = 1.0 + \gamma * V(s_i')$ and $s_i$ is the next state after action $a_1$. This update pushes the value of $Q(s_i, a_1)$ close to $1.0 + \gamma V(s_0')$. However, at the state $s_i$, we did not observe other actions, as it is usually the case in real world. Hence, there is no regressing target for $Q(s_i, a_0)$ and $Q(s_i, a_2)$. When we parametrize the Q function with a neural network, there is no guarantee of how $Q(s_i, a_0)$ and $Q(s_i, a_2)$ will compare to $Q(s_i, a_1)$. As a result, the learning algorithm cannot give meaningful Q values for the unobserved actions. The results in Figure 2 (left) has confirmed our analysis. A similar analysis applies to the hard Q method as well.

However, in the NAC method, the policy gradient update part is $(\nabla Q(s_i, a_1) - \nabla V(s_i))(Q(s_i, a_1) - \hat{Q}(s_i, a_1))$, where $V(s_i) = \log \sum_a exp(Q(s_i, a))$. After manipulating the terms, the update is equal to $\nabla \frac{exp(Q(s_i, a_1))}{\sum_a exp(Q(s_i, a))}(Q(s_i, a_1) - \hat{Q}(s_i, a_1))$. Note that the first term becomes the gradient of a cross entropy loss between a softmax layer output and a synthetic label of action $a_1$. The logits are $Q(s_i, \cdot)$. Intuitively, if the $Q(s_i, a_1)$ is under-estimated, the method tries to increase $Q(s_i, a_1)$ while pushing down $Q(s_i, a_0)$ and $Q(s_i, a_2)$. When $Q(s_i, a_1)$ is over-estimated, the reverse process happens. It is worth to note that the proposed method naturally has a prior of reducing the unobserved $Q(s_i, a_0)$ and $Q(s_i, a_2)$ values, when the current action $a_1$ is good.

## 6 CONCLUSION

We proposed a Normalized Actor-Critic algorithm for reinforcement learning from demonstrations. Our algorithm provides a unified approach for learning from reward and demonstration, and is robust to potentially suboptimal demonstration data. An agent can be fine-tuned with rewards after training on demonstrations by simply continuing to perform the same algorithm on on-policy data. Our algorithm preserves and then improves the behaviors learned from demonstration while receiving reward through interaction with an environment. Based on our experiments with the Torcs and GTA environments, we show that our algorithm is able to learn effectively and is robust to the arbitrary quality of demonstrations.

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

# Appendices

## A APPENDIX

### A.1 NORMALIZED ACTOR-CRITIC WITH Q PARAMETRIZATION

Usually the actor-critic method parametrizes $\pi(s, a)$ and $V(s)$ with a neural network that has two heads. In this section, we explore an alternative parametrization: Q-parametrization. Instead of outputting $\pi$ and $V$ directly, the neural network computes $Q(s, a)$. We parametrize $\pi$ and $V$ based on $Q$ by specifying a fixed mathematical transform:

$$V_Q(s) = \alpha \log \sum_a \exp(Q(s, a)/\alpha) \tag{13}$$

$$\pi_Q(a|s) = \exp((Q(s, a) - V_Q(s))/\alpha) \tag{14}$$

Note that the Q-parametrization we propose here can be seen as a specific design of the network architecture. Instead of allowing the net to output arbitrary $\pi(s, a)$ and $V(s)$ values, we restrict the network to only output $\pi(s, a)$ and $V(s)$ pairs that satisfy the above relationship. This extra restriction will not harm the network's ability to learn since the above relationship has to be satisfied at the optimal solution(Schulman et al., 2017; Haarnoja et al., 2017b; Nachum et al., 2017).

Based on the Q-parametrization, we can derive the update of the actor. Note that we assume the behavioral policy is $\pi_Q$, and we sample one step out of a trajectory, thus dropping the subscript $t$. The goal is to maximize expected future reward, thus taking gradient of it we get:

$$
\begin{aligned}
&\nabla \mathbb{E}_{s, a \sim \pi_Q} \left[ R(s, a) \right] \\
=&\mathbb{E}_{s, a \sim \pi_Q} \left[ R(s, a) \nabla \log_\theta p(a, s | \pi_Q) \right] \\
\approx&\mathbb{E}_{s, a \sim \pi_Q} \left[ R(s, a) \nabla_\theta \log \pi_Q(a|s) \right]
\end{aligned}
\tag{15}
$$

where the last step ignores the state distribution, thus an approximation. By adding some baseline functions, it turns to the following format, where $\hat{Q}(s, a) = R(s, a) + \gamma V_Q(s')$:

$$
\begin{aligned}
&\mathbb{E}_{s, a} \left[ \nabla_\theta \log \pi_Q(a|s)(\hat{Q}(s, a) - b(s)) \right] \\
=&\mathbb{E}_s \left[ \sum_a \pi_Q(a|s) \nabla_\theta \log \pi_Q(a|s)(\hat{Q}(s, a) - b(s)) \right]
\end{aligned}
\tag{16}
$$

As in previous work, an entropy-regularized policy gradient simply add the gradient of the entropy of the current policy with a tunable parameter $\alpha$ in order to encourage exploration. The entropy term is:

$$
\begin{aligned}
&\mathbb{E}_s \left[ \alpha \nabla_\theta H(\pi_Q(\cdot|s)) \right] \\
=&\mathbb{E}_s \left[ \alpha \nabla_\theta \sum_a -\pi_Q(a|s) \log \pi_Q(a|s) \right] \\
=&\mathbb{E}_s \left[ \alpha \sum_a -\nabla_\theta \pi_Q(a|s) \log \pi_Q(a|s) - \pi_Q(a|s) \nabla_\theta \log \pi_Q(a|s) \right] \\
=&\mathbb{E}_s \left[ \alpha \sum_a -\nabla_\theta \pi_Q(a|s) \log \pi_Q(a|s) - \pi_Q(a|s) \frac{1}{\pi_Q(a|s)} \nabla_\theta \pi_Q(a|s) \right] \\
=&\mathbb{E}_s \left[ \alpha \sum_a -\nabla_\theta \pi_Q(a|s) \log \pi_Q(a|s) \right] \\
=&\mathbb{E}_s \left[ \alpha \sum_a -\pi_Q(a|s) \nabla_\theta \log \pi_Q(a|s) \log \pi_Q(a|s) \right]
\end{aligned}
\tag{17}
$$

putting the two terms together and using the energy-based policy formulation (Eq. (14)) :

$$
\begin{aligned}
&\mathbb{E}_{s,a} \left[ \nabla_\theta \log \pi_Q(a|s)(\hat{Q}(s,a) - b(s)) + \alpha \nabla_\theta H(\pi_Q(\cdot|s)) \right] \\
=&\mathbb{E}_s \left[ \sum_a \pi_Q(s,a) \nabla_\theta \log \pi_Q(s,a)(\hat{Q}(s,a) - b(s) - (Q(s,a) - V_Q(s))) \right]
\end{aligned}
\tag{18}
$$

If we let the baseline $b(s)$ be $V_Q(s)$, we get the update:

$$
\begin{aligned}
&\mathbb{E}_s \left[ \sum_a \pi_Q(s,a) \nabla_\theta \log \pi_Q(s,a)(\hat{Q}(s,a) - Q(s,a)) \right] \\
=&\frac{1}{\alpha} \mathbb{E}_{s,a} \left[ (\nabla_\theta Q(s,a) - \nabla_\theta V_Q(s))(\hat{Q}(s,a) - Q(s,a)) \right]
\end{aligned}
\tag{19}
$$

where $\hat{Q}(s,a)$ could be obtained through bootstrapping by $R(s,a) + \gamma V_Q(s')$. In practice $\hat{Q}(s,a)$ is a target Q value and $V_Q(s')$ can be obtained from Eq. 13 and the parameters of the target network. For the critic, the update is:

$$
\begin{aligned}
&\mathbb{E}_s \left[ \nabla_\theta \frac{1}{2}(V_Q(s) - \hat{V}(s))^2 \right] \\
=&\mathbb{E}_s \left[ \nabla_\theta V_Q(s)(V_Q(s) - \hat{V}(s)) \right]
\end{aligned}
\tag{20}
$$

where $\hat{V}(s)$ could be obtained by $\hat{V}(s) = \mathbb{E}_a \left[ R(s,a) + \gamma V_Q(s') \right] + \alpha H(\pi_Q(\cdot|s))$.

## B  EFFECTS OF IMPERFECT DEMONSTRATIONS

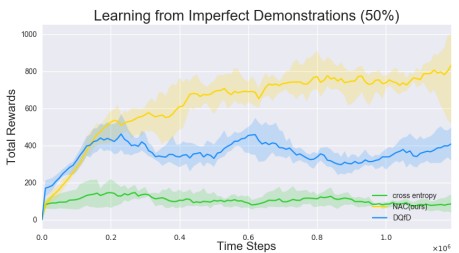 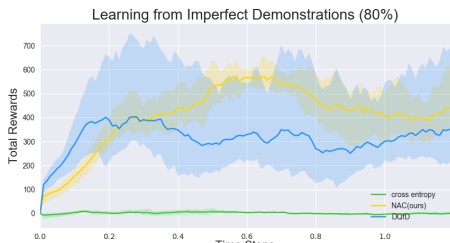

Figure 5: More results when introducing imperfect demonstrations. Left figure shows when there are 50% imperfect actions and the right one shows the case for 80%. Our NAC method is highly robust to noisy demonstrations.

## C  EFFECTS OF DEMONSTRATION AMOUNT

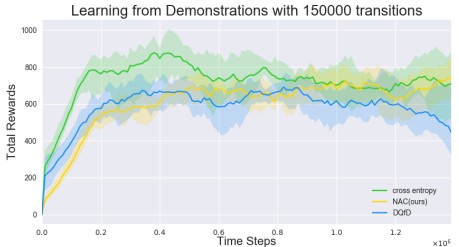 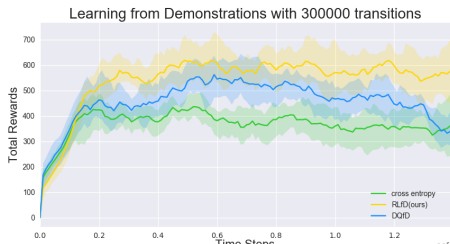

Figure 6: More results when varying the amount of demonstrations. The left and right figures show when there are 150k and 300k transitions respectively. Our NAC method achieves superior performance with a large amount of demonstrations and is comparable to supervise methods with smaller amount of demonstrations.

## D  EXPERIMENT DETAILS

**Network Architecture:** We use the same architecture as in Mnih et al. (2015) to parametrize $Q(s, a)$. With this Q parametrization, we also output $\pi_Q(a|s)$ and $V_Q(s)$ based on Eq. (6) and Eq. (5).

**Hyper-parameters:** We use a replay buffer with a capacity of 1 million steps and update the target network every 10K steps. Initially, the learning rate is linearly annealed from 1e-4 to 5e-5 for the first $1/10$ of the training process and then it is kept as a constant (5e-5). Gradients are clipped at 10 to reduce training variance. The reward discount factor $\gamma$ is set to 0.99. We concatenate the 4 most recent frames as the input to the neural network. For the methods with an entropy regularizer, we set $\alpha$ to 0.1, following Schulman et al. (2017). We truncate the importance sampling weighting factor $\beta = \min\left\{\frac{\pi_Q(a|s)}{\mu(a|s)}, c\right\}$ at 10, i.e., $c = 10$.

