# OpenReview forum: "Reinforcement Learning from Imperfect Demonstrations"
_ICLR.cc/2018/Conference — Invite to Workshop Track_

### Official Review · AnonReviewer2 · 2017-11-24
**New approach or new application?**

**Rating:** 5
**Confidence:** 3

**Review:**

Thanks for all the explanations on my review and the other comments. While I can now clearly see the contributions of the paper, the minimal revisions in the paper do not make the contributions clear yet (in my opinion that should already be clear after having read the introduction). The new section "intuitive analysis" is very nice.

*******************************

My problem with this paper that all the theoretical contributions / the new approach refer to 2 arXiv papers, what's then left is an application of that approach to learning form imperfect demonstrations.

Quality
======
The approach seems sound but the paper does not provide many details on the underlying approach. The application to learning from (partially adversarial) demonstrations is a cool idea but effectively is a very straightforward application based on the insight that the approach can handle truly off-policy samples. The experiments are OK but I would have liked a more thorough analysis.

Clarity
=====
The paper reads well, but it is not really clear what the claimed contribution is.

Originality
=========
The application seems original.

Significance
==========
Having an RL approach that can benefit from truly off-policy samples is highly relevant.

Pros and Cons
============
+ good results
+ interesting idea of using the algorithm for RLfD
- weak experiments for an application paper
- not clear what's new

---

> ### Public Comment · (anonymous) · 2018-01-05
> **Answer to your concerns**
>
> We thank you for the feedback and comments. We clarify our contributions in General Clarifications to Common Misunderstandings. Also, we answer the question about 1. Should demonstrations include reward signals?
> 2. Why NAC is not a simple application of an off-policy method to the learning from demonstration problem? in the General Clarifications.
>
> Besides the contributions that we respond to all reviewers, we would like to emphasize that although the technical details have appeared in [Haarnoja et al. 2017], our investigation of the method on the learning from demonstration problem and the better performance compared to the baselines are still novel. And as we explained in the common response, the NAC method is not simply taking an off-policy method and apply it on the learning from demonstration (LfD) problem. Many other off-policy methods such as Q learning doesn’t work at all on the LfD problem, while NAC has the correct prior that fits the LfD problems.
>
> The reviewer also suggests more extensive discussions and analysis of why our method perform better than baselines. We had some mathematical discussions in Section 3. To be more concrete, we pose an intuitive example here. Suppose at some state s0, we observe the demonstration has taken an action a1, and receiving a reward of 1.0. The action space is {a0, a1, a2} for every state. Soft Q takes an update of \nabla Q(s0, a1)*(Q(s0, a1) - \hat{Q(s0, a1)}) for the parameter \theta of the Q function, where \hat{Q(s0, a1)} = 1.0+gamma*V(s0’) and s0’ is the next state after action a1. This update pushes the value of Q(s0, a1) close to 1.0+gamma*V(s0’). However, at the state s0, we didn’t observe other actions, as it’s usually the case in real world, there is no regressing target for Q(s0, a0) and Q(s0, a2). When we parametrize the Q function with a neural network, there is no guarantee of how Q(s0, a0) and Q(s0, a2) will compare to Q(s0, a1). As a result, the learning algorithm won’t give any meaningful Q values for the unobserved actions. The results in Figure 2 (left) has confirmed our analysis. A similar analysis applies to the hard Q method as well.
>
> Now we turn to analyze the proposed NAC method. The policy gradient update part is (\nabla Q(s0, a1) - \nabla V(s0))*(Q(s0, a1) - \hat{Q(s0, a1)}), where V(s0) = log \sum_a exp(Q(s0,a)). After manipulating the terms, the update is equal to \nabla{ \frac {exp(Q(s0, a1))} {sum_a exp(Q(s0, a))} } * (Q(s0, a1) - \hat{Q(s0, a1)}). Note that the first term becomes the gradient of a cross entropy loss between a softmax layer output and a synthetic label of action a1. The logits are Q(s0, *). That is to say, when Q(s0, a1) - \hat{Q(s0, a1)} < 0, i.e. the Q(s0, a1) is under-estimated, the method tries to increase Q(s0, a1) while pushing down Q(s0, a0) and Q(s0, a2). When Q(s0, a1) is over-estimated, the reverse process happens. It’s worth to note that the proposed method naturally has a prior of reducing the undemonstrated Q(s0, a0) and Q(s0, a2) values, when the current action a1 is good enough, i.e. Q(s0, a1) is lower than the bootstrapped \hat{Q(s0, a1)} value. We also add the intuitive analysis in the newer version of our paper..
>
> DQfD works well when the demonstrated action is a good one. However, the supervise loss will deteriorate the performance when the actual demonstrated action is bad. The proposed NAC method, on the other hand, will learn from the bad demonstrations. In a case of the bad action, say the reward of doing a1 at state s0 is no longer 1.0, but -1.0 instead. The second term in the NAC update Q(s0, a1) - \hat{Q(s0, a1)} will more likely to turn positive, because the bootstrapped value is lower. In that case, the NAC will push down Q(s0, a1) and push up the other two values. Although there is no guarantee of the sign of the second term, it will definitely be more positive when the reward is lower. In contrast, DQfD does not have this adaptive behavior.
>
> Thank you for the comments on typos and references. We have already modified the texts and typos according to your suggestion.  [Haarnoja et al,. 2017] is already a published work at International Conference of Machine Learning (ICML) 2017 and PMLR.  We will correct the reference and cite the correct version.

---

### Official Review · AnonReviewer3 · 2017-11-26
**Similar to previous work - Experimental result not so convincing**

**Rating:** 6
**Confidence:** 5

**Review:**

This paper proposes a method to learn a control policy from both interactions with an environment and demonstrations. The method is inspired by the recent work on max entropy reinforcement learning and links between Q-learning and policy gradient methods. Especially the work builds upon the recent work by Haarnoja et al (2017) and Schulman et al (2017) (both unpublished Arxiv papers).

I'm also not sur to see much differences with the previous work by Haarnoja et al and Schulman et al. It uses demonstrations to learn in an off-policy manner as in these papers. Also, the fact that the importance sampling ration is always cut at 1 (or not used at all) is inherited from these papers too.

The authors say they compare to DQfD but the last version of this method makes use of prioritized replay so as to avoid reusing too much the expert transitions and overfit (L2 regularization is also used). It seems this has not been implemented for comparison and that overfitting may come from this method missing.

I'm also uncomfortable with the way most of the expert data are generated for experiments. Using data generated by a pre-trained network is usually not representative of what will happen in real life. Also, corrupting actions with noise in the replay buffer is not simulating correctly what would happen in reality. Indeed, a single error in some given state will often generate totally different trajectories and not affect a single transition. So imperfect demonstration have very typical distributions. I acknowledge that some real human demonstrations are used but there is not much about them and the experiment is very shortly described.

---

> ### Public Comment · (anonymous) · 2018-01-05
> **Answer to your concerns**
>
> We thank you for the feedback and comments. We clarify our contributions in General Clarifications to Common Misunderstandings.
>
> About the baseline DQfD, we have already included the L2 regularization loss in the current version of our experiments, and we will clarify it in the paper. We did not include the prioritized replay buffer in our re-implementation because the prioritized replay was not a DQfD component when we wrote the paper. Moreover, since prioritized replay is an independent component for both algorithms, adding it will likely have a similar effect on both NAC and DQfD. However, to be complete in comparison, we will add one more comparison with prioritized replay in the final version.
>
> The reviewer also mentions about the noisy data generation process. We generated data from a trained agent, but we did not simply add noise to the replay buffer. Instead, we corrupt the data while collecting them; the driving agent will have the compound error you mentioned in your comments. To quantify the quality of a dataset, we use a trained agent because we can measure the amount of corrupted actions and use that to indicate how imperfect the dataset is.
>
> We collected data from amateur human players with their natural capability of playing video games. We can intentionally let the players do wrong actions but it will not be significantly different from trained agents. We also detail more about the human data and our experiments on human dataset accordingly.
>
> Thank you for correcting the reference. [Haarnoja et al,. 2017] is already a published work at International Conference of Machine Learning (ICML) 2017 and PMLR.  We will correct the reference and cite the correct version.

---

### Official Review · AnonReviewer1 · 2017-11-28
**Unclear derivations; Novelty is not obvious**

**Rating:** 5
**Confidence:** 4

**Review:**

SUMMARY:

The motivation for this work is to have an RL algorithm that can use imperfect demonstrations to accelerate learning. The paper proposes an actor-critic algorithm, called Normalized Actor-Critic (NAC), based on the entropy-regularized formulation of RL, which is defined by adding the entropy of the policy as an additional term in the reward function.
Entropy-regularized formulation leads to nice relationships between the value function and the policy, and has been explored recently by many, including [Ziebart, 2010], [Schulman, 2017], [Nachum, 2017], and [Haarnoja, 2017].
The paper benefits from such a relationship and derives an actor-critic algorithm. Specifically, the paper only parametrizes the Q function, and computes the policy gradient using the relation between the policy and Q function (Appendix A.1).

Through a set of experiments, the paper shows the effectiveness of the method.


EVALUATION:

I think exploring and understanding entropy-regularized RL algorithm is important. It is also important to be able to benefit from off-policy data. I also find the empirical results encouraging. But I have some concerns about this paper:

- The derivations of the paper are unclear.
- The relation with other recent work in entropy-regularized RL should be expanded.
- The work is less about benefiting from demonstration data and more about using off-policy data.
- The algorithm that performs well is not the one that was actually derived.

* Unclear derivations:
The derivations of Appendix A.1 is unclear. It makes it difficult to verify the derivations.

To begin with, what is the loss function of which (9) and (10) are its gradients?

To be more specific, the choices of \hat{Q} in (15) and \hat{V} in (19) are not clear.  For example, just after (18) it is said that “\hat{Q} could be obtained through bootstrapping by R + gamma V_Q”. But if it is the case, shouldn’t we have a gradient of Q in (15) too? (or show that it can be ignored?)

It appears that \hat{Q} and \hat{V} are parameterized independently from Q (which is a function of theta). Later in the paper they are estimated using a target network, but this is not specified in the derivations.

The main problem boils down to the fact that the paper does not start from a loss function and compute all the gradients in a systematic way. Instead it starts from gradient terms, each of which seems to be from different papers, and then simplifies them. For example, the policy gradient in (8), which is further decomposed in Appendix A.1 as (15) and (16) and simplified, appears to be Eq. (50) of [Schulman et al., 2017] (https://arxiv.org/abs/1704.06440). In that paper we have Q_pi instead of \hat{Q} though.

I suggest that the authors start from a loss function and clearly derive all necessary steps.


* Unclear relation with other papers:
What part of the derivations of this work are novel? Currently the novelty is not obvious.
For example, having the gradient of both Q and V, as in (9), has been stated by [Haarnoja et al., 2017] (very similar formulation is developed in Appendix B of https://arxiv.org/abs/1702.08165).
An algorithm that can work with off-policy data has also been developed by [Nachum, 2017] (in the form of a Bellman residual minimization algorithm, as opposed to this work which essentially uses a Fitted Q-Iteration algorithm as the critic).

I think the paper could do a better job differentiating from those other papers.


* The claim that this paper is about learning from demonstration is a bit questionable. The paper essentially introduces a method to use off-policy data, which is of course important, but does not cover the important scenario where we only have access to (state,action) pairs given by an expert. Here it appears from the description of Algorithm 1 that the transitions in the demonstration data have the same semantic as the interaction data, i.e., (s,a,r,s’).
This makes it different from the work by [Kim et al., 2013], [Piot et al., 2014], and [Chemali et al., 2015], which do not require such a restriction on the demonstration data.


* The paper mentions that to formalize the method as a policy gradient one, importance sampling should be used (the paragraph after (12)), but the performance of such a formulation is bad, as depicted in Figure 2. As a result, Algorithm 1 does not use importance sampling.
This basically suggests that by ignoring the fact that the data is collected off-policy, and treating it as an on-policy data, the agent might perform better. This is an interesting phenomenon and deservers further study, as currently doing the “wrong” things is better than doing the “right” thing. I think a good paper should investigate this fact more.

---

> ### Public Comment · (anonymous) · 2018-01-05
> **Answer to your concerns**
>
> We thank you for the feedback and comments. We clarify our contributions in General Clarifications to Common Misunderstandings. Also, we answer the question about 1. Should demonstrations include reward signals?
> 2. Why NAC is not a simple application of an off-policy method to the learning from demonstration problem? in the General Clarifications.
>
> We clarify our derivation of our method in Appendix. A. We reorganize Appendix A to derive the loss and gradient from a single objective.
>
> In the newer version of the paper, we also include the comparison with the PCL method [Nachum et al., 2017]. We only include this method in the first set of experiment and find that our method consistently outperforms the baseline. We also plan to include this baseline in other settings methods in the final version of this paper.
>
> The reviewer also suggested the fact that NAC without important sampling is better than NAC with it deserves further study. We planned to study this phenomenon in depth before the final version. Specifically, we want to test the hypothesis that it is the high variance of the importance sampling gradient estimator that causes the instability. Specifically, at some parameter \theta_0, we could calculate the gradient either with importance sampling, or without it. By repeatedly compute the gradient with different data mini-batch at the same \theta_0 for a sufficient number of rounds, we could get two gradient estimators’ bias and variance respectively. With those statistics, one could compare the two convergence speeds with the theories of stochastic gradient descent. We also plan to incorporate the ideas from [Gu et al. 2017] to reach a better policy gradient estimator by combining the two methods.
>
> [1] Shixiang Gu, Timothy Lillicrap, Zoubin Ghahramani, Richard E. Turner, Bernhard Schölkopf, Sergey Levine. “Interpolated Policy Gradient: Merging On-Policy and Off-Policy Gradient Estimation for Deep Reinforcement Learning”. NIPS 2017.

---

### Comment · AnonReviewer2 · 2017-11-24
**So what's new?**

The paper proposes to employ an algorithm from Haarnoja 2017 and Schulman 2017 for reinforcement learning where some of the data comes from potentially adversarial user demonstrations.
The paper does not seem to be able to make up its mind whether the approach is novel or not. "We propose a unified LfD approach" but then all the algorithmic details are "as shown by Haarnoja / Schulman 2017". It gets a bit messy with arXiv papers (in this case they don't seem to have appeared yet in a peer-reviewed venue) but the authors are treating these two papers as published, so I'll also treat this paper as an extension of these. Which means that this paper boils down to "just" taking an RL approach from Haarnoja / Schulman 2017 that works well with data that is truly off-policy and employing it for learning from (imperfect) demonstrations. Hence we have essentially an experimental paper.
The approach does reasonably well, but I'd like to have seen more extensive discussions and an analysis WHY it performs better (or in some experiments actually worse) compared to the baselines.

Minor comments
===============
Eq (1): \operatorname{argmax} https://en.wikibooks.org/wiki/LaTeX/Advanced_Mathematics
Eq (4): I found this very strange, the text reads like this result is from 2010 while the entropy formulation above is claimed to be from 2017...
Sect. 3: "Equation 8" -> "Equation (8)" \eqref{}
Sect. 3.1: So what's new compared to Haarnoja / Schulman 2017?
Sect. 3: you talk about normalization but only explain in Sect 3.2 (and even there the discussion should be improved)  which part of the update corresponds to this normalization
Sect. 5.2: "method(Mnih" => "method (Mnih"

---

### Public Comment · (anonymous) · 2018-01-05
**General Clarifications to Common Misunderstandings**

Here we further differentiate our work from related ones. Both of [Haarnoja et al., 2017] and [Schulman et al., 2017] focus on soft Q learning, where the update is derived from L2 loss of the Bellman error. The soft Q formulation in [Haarnoja et al., 2017] and [Schulman et al., 2017] are exactly the same when the action space is discrete and we present it in our experiments as the "soft Q" method. Our proposed NAC algorithm can only be derived when we start from a soft policy gradient perspective, where the objective is the future expected discounted sum of rewards. As we’ve shown in the paper, the \nabla V term is the key difference between soft Q and NAC, and that extra term normalizes the Q value. We have a similar formulation as the one described in Appendix. B. in [Haarnoja et al., 2017]. However, they didn’t implement the algorithm, not to say fully explore the properties of such a NAC method. It’s because the method doesn’t have any obvious benefit as an off-policy learning method. We discover that in the learning from demonstration setting, the NAC method is quite useful and outperforms all the baselines.

We summarize by restating our novelties and contributions as follows:
1. 	We are the first to explore the NAC method experimentally in the learning from demonstration scenario and discovered that the proposed method is not only theoretically plausible but also outperforms alternative methods experimentally.
2. 	To the best of our knowledge, we are the first to propose a unified method to learn from both demonstration and environment interaction.
3. 	Unlike other methods that utilize supervised learning to learn from demonstration, our pure reinforcement learning method is not sensitive to noisy demonstrations and it outperforms prior methods that explicitly mix supervised learning and reinforcement learning objectives.

Reviewer 1 and 2 also mention that the proposed method is more about using off-policy data, rather than using demonstrations. We would like to clarify two aspects here:

1. Should demonstrations include reward signals?
2. Why NAC is not a simple application of an off-policy method to the learning from demonstration problem?

For the first question, some literatures take a demonstration set as a collection of (s, a) pairs, such as [Kim et al., 2013] mentioned by Reviewer 1. While other works, such as [Hester et al., 2017], also include the reward into the demonstrations, i.e. the (s, a, r) tuple. Learning from demonstration is a broad concept and we refer to the second setting in this paper. We also clarify this in the related work section of the paper.

For the second question, we first note the difference between the demonstration and the general off-policy data. Although one could view the demonstration set in our paper as some off-policy data, they are critically different because demonstrations are mostly good behaviors while that is not necessarily true for off-policy data. That’s why generic off-policy methods such as Q learning and soft Q learning don’t work on demonstration set at all, as described in Section 3 of the paper. Our method, on the other hand, has the normalization factor that causes the method to mimic the demonstration data when there is no evidence to the contrary. This implies an assumption that the prior off-policy data is generally good, or at least better than a random policy. This is a major conceptual distinction from Q-learning and soft Q-learning methods and, indeed, as shown in our experiments, our method substantially outperforms these prior methods.

---

### Decision · Program_Chairs · 2018-01-29
**ICLR 2018 Conference Acceptance Decision**

**Decision:**

Invite to Workshop Track

**Comment:**

I appreciate the experimental results, which includes a comparison against several baselines, however, I echo some of the concerns raised by the reviewers that the formulation is unclear and hard to follow. Moreover, the novelty over [Nachum, 2017] and [Haarnoja, 2017] seems small. Especially because [Nachum, 2017] also used expert trajectories to improve the performance in their experiments.

Detailed comment:
The use of log-sum-exp state values is only valid for the optimal policy, so it is not clear how an on-policy state value is replaced with the log-sum-exp state value. Also, because the equations that you derive characterize the optimal policy, I am not sure if you need importance correction at all.